# Nanoarchitectonics of Three-Dimensional Carbon Nanofiber-Supported Hollow Copper Sulfide Spheres for Asymmetric Supercapacitor Applications

**DOI:** 10.3390/ijms24119685

**Published:** 2023-06-02

**Authors:** Miyeon Shin, Ganesh Prasad Awasthi, Krishna Prasad Sharma, Puran Pandey, Mira Park, Gunendra Prasad Ojha, Changho Yu

**Affiliations:** 1Department of Energy Storage, Conversion Engineering of Graduate School, Jeonbuk National University, Jeonju 54896, Republic of Korea; 2Division of Convergence Technology Engineering, Jeonbuk National University, Jeonju 54896, Republic of Korea; 3Division of Physics and Semiconductor Science, Dongguk University, Seoul 04620, Republic of Korea; 4Carbon Composite Energy Nanomaterials Research Center, Woosuk University, Wanju 55338, Republic of Korea

**Keywords:** hollow copper sulfide, 3D-carbon nanofibers, HCuS@3D-CNF nanoarchitectonics, asymmetric supercapacitors

## Abstract

Three-dimensional carbon nanofiber (3D-CNF)-supported hollow copper sulfide (HCuS) spheres were synthesized by the facile hydrothermal method. The morphology of the as-synthesized HCuS@3D-CNF composite clearly revealed that the 3D-CNFs act as a basement for HCuS spheres. The electrochemical performance of as-synthesized HCuS@3D-CNFs was evaluated by cyclic voltammetry (CV) tests, gravimetric charge–discharge (GCD) tests, and Nyquist plots. The obtained results revealed that the HCuS@3D-CNFs exhibited greater areal capacitance (4.6 F/cm^2^) compared to bare HCuS (0.64 F/cm^2^) at a current density of 2 mA/cm^2^. Furthermore, HCuS@3D-CNFs retained excellent cyclic stability of 83.2% after 5000 cycles. The assembled asymmetric device (HCuS@3D-CNFs//BAC) exhibits an energy density of 0.15 mWh/cm^2^ with a working potential window of 1.5 V in KOH electrolyte. The obtained results demonstrate that HZnS@3D-CNF nanoarchitectonics is a potential electrode material for supercapacitor applications.

## 1. Introduction

Recent research has been focused on next-generation energy storage devices to meet the demands of portable electronics and digital communications [1,2]. Supercapacitors (SCs), also known as ultracapacitors, are considered promising candidates for energy storage by virtue of their advantages such as high power density, thousands of charge–discharge cycles, and operational durability [3,4,5]. However, maintaining the energy density, power density, and lifecycle of SCs is challenging in practical applications [6,7]. Therefore, hybrid supercapacitors must be designed to have higher specific capacitance without losing their essential characteristics.

In the past few years, various pseudo-capacitive materials such as transition metal oxides/or hydroxides [8,9,10,11,12,13,14,15], transitional metal sulfides (TMSs) [16,17,18,19,20,21], and conductive polymers [22,23] have been extensively studied as auspicious electrodes for supercapacitor applications. TMSs have received growing attention for supercapacitor electrode applications due to their controlled shapes, low electronegativity, suitable potential window, better electrical conductivity compared to their corresponding oxides, ample electrochemical redox sites, low cost, and environmental geniality, and because of the complex valence of sulfur [24,25,26,27]. Among various TMSs, copper sulfides [28] are considered p-type semiconductors and have become a predominant SC electrode material due to their abundant availability, easy approachability, and large theoretical capacity. However, they have more resistance to electron and/or ion transfer, low rate capability, and low cyclic durability, factors that limit their potential to be applied in electrode materials for supercapacitors [29,30]. In order to overcome these limitations, the rational design of CuS with suitable nanoarchitectures can expand opportunities to optimize their electrochemical performance.

Well-known carbonaceous materials such as graphene, activated carbon, carbon nanotubes, and carbon nanofibers have a large surface area, high conductivity, flexibility, and chemical and thermal stability, and can act as surface stabilizers for nanomaterials and improve the electron transfer among nanoparticles [31,32]. Very few reports have been published that combine carbon-based materials with different morphologies of CuS for supercapacitor applications. Recently, Niu et al. synthesized MOF-derived copper sulfide polyhedrons in carbon nanotube networks for hybrid supercapacitors [33], Hout S.E. et al. synthesized reduced graphene oxide-anchored hexagonal copper sulfide nanospheres [34], and Sabeeh H. et al. synthesized CuS nanochips with carbon nanotubes [35]. Huang K.J. et al. synthesized acetylene black-incorporated layered CuS nanosheets for high-performance supercapacitors [36]. Zhao et al. synthesized a 3D CuS@CD-CNT composite as a supercapacitor electrode material [37]. Similarly, Wang et al. synthesized a CuS nanoflower-decorated activated carbon layer for asymmetric supercapacitors [38]. In recent years, 3D-structured CNFs have gained much attention in supercapacitor applications due to their noteworthy characteristics such as easy synthesis, light weight, numerous conductive paths, and high electrolyte ion accessibility and diffusion during electrochemical tests. Recently, Tiwari et al. demonstrated that the Co@3D-CNF composite provides improved electrochemical performance compared to the Co@2D-CNF composite for supercapacitor applications [39]. Similarly, Dahal et al. fabricated MOF-assimilated B/N co-doped 3D-CNFs for binder-free electrodes for supercapacitors [40].

Considering the above studies, the HCuS@3D-CNF composite was synthesized by converting two-dimensional (2D) PAN electrospun nanofibers into 3D-CNFs, followed by the hydrothermal method. In HCuS@3D-CNF nanoarchitectonics, the loosely packed 3D-CNFs provide a conductive basement with a large surface area and free space to accommodate possible volumetric changes during charging and discharging processes. As a result, composite materials allow efficient electron transfer for enhanced specific capacitance, conductivity, and cyclic stability compared to bare HCuS. Moreover, an asymmetric supercapacitor [41] device employing HCuS@3D-CNF as a cathode and biomass-derived activated carbon [42] as a negative electrode was assembled to analyze the energy density and power density for practical applications. This kind of 3D fibrous structured carbon-based composite can be new avenue for possible energy storage applications.

## 2. Results and Discussion

### 2.1. Physicochemical Studies

Figure 1 shows the surface and structural morphology of the as-synthesized HCuS spheres and the HCuS@3D-CNF composite. Figure 1a,b show the polycrystalline hollow CuS nanospheres at low and high magnification, respectively. The surface of the HCuS spheres was created with numerous nanoflakes. When the as-prepared 3D-CNFs were added to the CuS precursors by the hydrothermal method, HCuS spheres were scattered throughout the surface of the 3D-CNFs (Figure 1c,d). Figure 1c,d are low- and high-magnification images of the HCuS@3D-CNF composite, respectively. Additionally, the surface diameter of HCuS spheres was considerably decreased compared to pristine HCuS spheres due to the increasing active edge sites present on the surface of 3D-CNFs. The active edge sites of the 3D-CNFs provide an easy way for the growth and nucleation of HCuS crystal [43]. The elements existing in the HCuS@3D-CNF composite were confirmed by EDS and elemental mapping (Figure 1e–j). The results clearly show the presence of Cu, S, and C throughout the scanned area.

Corresponding TEM images of the as-synthesized composite are displayed in Figure 2. Figure 2a shows that the hollow spheres of CuS were well attached to the surface of 3D-CNFs under the hydrothermal process, which is appropriate with FESEM morphology. Moreover, Figure 2b displays the HRTEM image of HCuS@3D-CNFs with the lattice fringes of the interplanar spacing of hexagonal CuS of 0.32 nm corresponding to the crystal plane (100) with amorphous 3D-CNFs, which suggests the intimate contact of CuS and the carbon nanofibers [44]. The result of the SAED pattern reveals that the polycrystalline nature of the CuS@3D-CNF composite was formed.

Figure 3 shows the XRD patterns and Raman spectra of HCuS, 3D-CNFs, and the HCuS@3D-CNF composite. In Figure 3a, the diffraction peaks at different phase angles of 2θ = 10.7°, 27.7°, 29.43°, 31.8°, 32.9°, 38.9°, 43.1°, 44.3°, 48.01°, 52.7°, 56.2°, 57.3°, 59.3°, 63.5°, 67.4°, 69.6°, 74.1°, 77.8°, and 79.2° with their corresponding planes (002), (101), (102), (103), (006), (105), (106), (008), (110), (108), (201), (202), (116), (1010), (118), (1011), (208), (212), and (213), respectively, indicate the hexagonal phase of CuS. The characteristic peaks of CuS are in good agreement with JCPDS No. 06-0464 [45,46]. The XRD pattern of pristine 3D-CNFs shows a broad diffraction peak at around 24°, corresponding to the crystal plane (002), which represents hexagonal graphitic carbon with an amorphous nature. Furthermore, the XRD patterns of the HCuS@3D-CNF composite match well with pristine HCuS along with 3D-CNFs, attributed to the composite formation. In addition, lower peak intensities compared to pristine HCuS demonstrate the efficacious incorporation of HCuS@3D-CNFs. The Raman spectra of different samples are shown in Figure 3b. The Raman spectra of HCuS exhibited two district peaks at about 270 cm^−1^ and 474 cm^−1^, belonging to Cu-S bond vibration and S-S covalent bond of stretching mode [47]. The Raman shift of 3D-CNFs showed characteristic broad bands at around 1343 cm^−1^, indicating the D band (C=C disordered and crystal defect of A_1g_ symmetry), and the broad band around 1577 cm^−1^ represents the D band (Sp^2^ graphitic lattice vibration mode with E_2g_ symmetry) [48,49,50]. Furthermore, the lower-intensity peaks of Cu-S and S-S bands along with D and G bands of 3D-CNFs represent the HCuS crystal growth on the surface of 3D-CNFs in the HCuS@3D-CNF composite. The obtained results of Raman spectra are in good agreement with XRD patterns.

The elemental composition and electronic structure of the individual elements in the HCuS@3D-CNFs were evaluated by XPS, as shown in Figure 4. Figure 4a shows the binding energy of Cu (932.9 eV), S (164.11eV), and C (285.5 eV) in the wide-range XPS survey spectrum. Furthermore, the core levels of XPS spectra of Cu 2p, S 2p, and C 1s are presented in Figure 4b–d, respectively. In the case of Cu 2p, two binding energies at 933.1 eV and 953.2 eV conform to Cu 2p^3/2^ and Cu 2p^1/2^, respectively. In the high-resolution S 2p spectrum, peaks of binding energies at about 162.9 eV and 164.08 eV correspond to S 2p^3/2^ and S 2p^1/2^, respectively. In addition, a lower-intensity broad peak at 168.5 eV represents the sulfate molecules (SO_4_^2−^) due to the adventitious surface oxidation of the heterostructure. The peaks of 2p^3/2^ and 2 p^1/2^ of Cu 2p and S 2p are attributed to Cu^2+^ and S^2−^ states, revealing the CuS formation. The high-resolution spectra of C 1s exhibit an intense peak at about 285 eV, corresponding to a C-C bond, and a weak peak at about 289 eV, corresponding to a C-O-C bond. The oxygen atom contained with carbon in C 1s might be triggered by the atmosphere [51].

### 2.2. Electrochemical Studies

In order to evaluate the electrochemical performances of HCuS and the HCuS@3D-CNF composite for supercapacitors, cyclic voltammetry (CV), galvanostatic charge–discharge (GCD), electrochemical impedance spectroscopy, and cyclic stability tests were performed with aqueous 2 M KOH electrolyte in a three-electrode system (Figure 5). The CV tests of HCuS and the HCuS@3D-CNF composite were carried out at various scan rates (10 to 100 mV) with a potential window of 0–0.5 V, as shown in Figure 5a,b. The CV curves of the HCuS@3D-CNF composite have a larger enclosed area under CV curves along with a higher current response compared to HCuS. The larger enclosed area of the heterostructure is due to the synergistic effect between HCuS and 3D-CNFs. The synergistic effect could be responsible for the shorter diffusion path for electrolyte ion transport onto the HCuS@3D-CNF electrode surface. The CV curves of both samples (HCuS and HCuS@3D-CNF) exhibited pseudo-capacitive behavior due to the Cu/Cu^2+^ redox in the KOH electrolyte [52,53]. The gravimetric charge–discharge behavior of the samples was analyzed from the GCD curve at various current densities (1 to 10 mA/cm^2^), as shown in Figure 5c,d. The HCuS@3D-CNFs showed noteworthy discharge times compared to HCuS. Based on discharge time, the areal capacitances of the HCuS@3D-CNF electrodes are 4.65 F/cm^2^, 4.47 F/cm^2^, 4.36 F/cm^2^, 4.0 F/cm^2^, and 3.93 F/cm^2^ at the current densities of 2, 4, 6, 8, and 10 mA/cm^2^, respectively; the HCuS electrode had areal capacitances of 0.64 F/cm^2^, 0.64 F/cm^2^, 0.63 F/cm^2^, 0.63 F/cm^2^, and 0.62 F/cm^2^ at these same current densities. The areal capacitance of electrodes gradually decreased with increasing current densities. The inverse relation between capacitance and current density depends on the diffusion and migration of electrolyte ions. It is noticeable that the rate of ion diffusion and migration is lower at a higher current density, leading to a decrease in capacitance value. The higher capacitance of HCuS@3D-CNFs is due to the enhanced surface area provided by the loosely packed surface of the three-dimensional carbon nanofibrous morphology. In addition, the open pores and surface roughness of 3D-CNFs with HCuS provide better exposure for active electrolyte ions’ accessibility [54,55]. Regarding the electrochemical activity in the CuS and CuS/C composite electrodes, the following reactions take place during the oxidation/reduction processes [56].

Reduction process:CuS + OH^−^ → CuSOH + e^−^
(1)
CuSOH + OH^−^ → CuSO + H_2_O + e^−^
(2)

Oxidation process:CuSO + H_2_O + e^−^ → CuSOH + OH^−^
(3)
CuSOH + e^−^ → CuS + OH^−^
(4)

The areal capacitances vs. current densities of HCuS and the HCuS@3D-CNF composite were evaluated (Figure 6a). During the GCD test, the gradual decrease in discharge time with an increase in the current density (2–10 mA/cm^2^) was due to the internal resistances and kinetically slow redox reactions [53]. The EIS Nyquist plot of HCuS and the HCuS@3D-CNF composite at a circuit potential of 10 mV in the frequency range of 0.01 Hz to 100 KHz is shown in Figure 6b. The negligible semicircle in the higher-frequency region followed by the more inclined line in the lower-frequency region of HCuS@3D-CNFs compared to HCuS indicates fast electron transport and a shortened diffusion path for electrolyte ions. The equivalent circuit model of the Nyquist plot represents the charge transfer resistance (Rct) and Warburg resistance (W) at the working electrode’s surface due to faradic and ELDC behavior at the electrode surface and the slope of the straight line in the lower-frequency region, respectively. Rs represents the bulk resistance in the high-frequency region. The cyclic stability test of the HCuS@3D-CNF composite was performed at 10 A/g current density (Figure 6c). The results suggest that the capacitance retention of ~95% was maintained after 5000 cycles. The GCD curves before and after the cyclic test are depicted in the inset of the figure.

#### Asymmetric Supercapacitor

For the fabrication of an asymmetric supercapacitor, biomass-derived activated carbon [42] was selected for the negative electrode due to its large surface area, porosity, and low cost. Figure 7a shows the schematic illustration of the assembled electrochemical cell for the asymmetric supercapacitor (HCuS@3D-CNFs//BAC). As the first stage in the asymmetric electrochemical studies, the individual cyclic voltammograms were operated for positive and negative electrodes under a three-electrode system. For the negative electrode, the CV, GCD, and EIS curves are given in Appendix A. The CV curves of BAC reveal the EDLC behaviors at various scan rates of 10–100 mV/s, which occur due to the electrostatic adsorption of electrolyte ions on the surface of the electrode. The symmetrical-shaped GCD curves exhibit areal capacitances of 0.255, 0.218, 0.19, 0.16, 0.15, and 0.08 F/cm^2^ at 2, 4, 6, 8, 10, and 20 mA/cm^2^ current densities, respectively. For the EIS measurement, the semicircle in the high-frequency region and the linear steep slope in the medium- and low-frequency regions establish its conductive nature [57]. For the estimation of the potential window, the optimal CV curve of the positive electrode (HCuS@3D-CNFs) was analyzed over the potential window of 0–0.5 V, and for the negative electrode, the window of −1 to 0 V was used [42] at the same scan rate (50 mV/s) (Figure 7b). After assembling both materials, the maximum working potential window of the CV was expanded to 0–1.5 V due to the synergistic effect of cathode and anode materials [58]. The CV curve of the assembled cell was analyzed at various scan rates (10–100 mV/s) (Figure 7c). The synergistic effect can be observed in the shape of the CV curve, as shown in Figure 7d. The EDLC behavior is revealed until the potential of 1.2 V, whereas the redox peaks are demonstrated above the potential of 1.2 V at 50 mV/s. The faradic and non-faradic behaviors revealed in the voltammogram suggest that the assembled asymmetric supercapacitor using BAC as the negative electrode and the HCuS@3D-CNF composite as the positive electrode can be operated up to 1.5 V in KOH electrolyte [59]. Moreover, there was a mild shift in the peak position at a higher scan rate owing to the polarization behavior of the electrode [60]. The GCD curves of the asymmetric device measured over the potential window of 0–1.5 V at various current densities are shown in Figure 7e. The symmetrical GCD curves of HCuS@3D-CNFs//AC demonstrate pseudo-capacitive behaviors with remarkable areal capacitances of 0.49 F/cm^2^, 0.43 F/cm^2^, 0.37 F/cm^2^, 0.33 F/cm^2^, and 0.30 F/cm^2^ at various current densities of 2 mA/cm^2^, 4 mA/cm^2^, 6 mA/cm^2^, 8 mA/cm^2^, and 10 mA/cm^2^, respectively. To evaluate the applicability of the synthesized material, the energy density (E) and power density (P) of asymmetric supercapacitors are the key parameters in a wide potential range. Therefore, energy density and power density were evaluated using Equations (3) and (4), respectively. Based on the different current densities (2–10 mA/cm^2^), we obtained energy densities of 0.15 mWh/cm^2^, 0.13 mWh/cm^2^, 0.11 mWh/cm^2^, and 0.10 mWh/cm^2^ corresponding to power densities of 2.92 mW/cm^2^, 5.74 mW/cm^2^, 8.50 mW/cm^2^, 11.45 mW/cm^2^, and 14.09 mW/cm^2^, respectively. The obtained values are superior to the previous reports (Table 1). Moreover, the charge–discharge curves of the asymmetric device were measured over different potential windows at a current density of 2 mA/cm^2^ (Figure 7f). The symmetrical shape of all GCD curves reveals the remarkable capacitive performance.

Cyclic stability is one another important parameter to estimate the durability of a supercapacitor electrode. The cyclic stability of the device (HCuS@3D-CNFs//BAC) was evaluated by the cyclic stability test (5000 cycles @ 10 mA/cm^2^) (Figure 8). The cyclic stability results showed that the device maintained 83.2% capacitance retention after 5000 cycles (Figure 8a). The Nyquist plot of EIS of the asymmetric device before and after the cyclic stability test is given in Figure 8b. The obtained results reveal a negligible semicircle in the high-frequency region and a comparatively lower vertical line in the lower-frequency region after 5000 cycles, which indicates a slight decrease in the conductivity of the device.

## 3. Methods and Materials

### 3.1. Materials

Polyacrylonitrile (PAN, Mw 150000, Sigma Aldrich, St. Louis, MO, USA), copper sulfate pentahydrate (CuSO_4_.5H_2_O; DaeJung chemicals and materials Co., Ltd., Busan, Korea), thiourea (Junsei chemical Co., Ltd., Tokyo, Japan), N-N dimethylformamide (DMF, 99.5% Sigma-Aldrich), and sodium borohydride (NaBH_4_, Sigma-Aldrich) were used in this experiment. All chemicals were of analytical grade and used without any further purification.

### 3.2. Sample Preparation

#### 3.2.1. Preparation of 3D-CNFs

The 3D-CNFs were synthesized according to our previous protocol [69,70,71]. Firstly, 10 wt % PAN-DMF solution was electrospun to obtain a PAN nanofibrous mat. The nanofibrous mat was dried under vacuum at 60 °C for 24 h. During the preparation of the 3D structure, the as-fabricated nanofibrous mat was placed into 20 mL of NaBH_4_ (0.1 M) solution. After 24 h, the 3D spongy mat was washed several times with deionized water and vacuum dried at 60 °C for 24 h, followed by carbonization at 950 °C for 2 h under N_2_ atmosphere after stabilization in air at 250 °C for 2 h.

#### 3.2.2. Preparation of HCuS@CNF nanoararchitectonics 

A schematic representation of the fabrication process of HCuS@3D-CNF composites is given in Figure 9. First, copper sulfate pentahydrate (1 mM) and thiourea (2.5 mM) were each dissolved in 20 mL distilled water separately. After sufficient dispersion, thiourea solution was transferred into the copper sulfate pentahydrate solution and magnetically stirred again for 30 min. A piece of as-synthesized 3D-CNF (0.02 g) was soaked in the mixed solution and transferred to a 100 mL Teflon vessel autoclave for the hydrothermal process. The oven temperature was maintained at 180 °C for 24 h. The obtained HCuS@CNF composite was washed successively with deionized water and ethanol and dried at 50 °C for 12 h. For comparison, bare HCuS spheres were prepared without 3D-CNFs.

### 3.3. Characterizations

The structural morphologies of the as-synthesized samples were analyzed by field emission scanning electron microscopy (FESEM, Hitachi, Tokyo, Japan). Furthermore, the internal morphology of the samples was evaluated by a high-resolution transmission electron microscope (HRTEM, JEOL Ltd., Tokyo, Japan) operated at 200 kV. The phase composition and crystal orientation of the samples were analyzed using high-resolution X-ray diffraction (HR-XRD, Rigaku Co., Tokyo, Japan). The degree of graphitization in carbon with CuS was studied with a high-resolution dispersive Raman microscope (RFS-100S, Bruker, Mannheim, Germany) with a laser source wavelength of 514 nm. The bonding configurations were determined by Fourier transform infrared spectroscopy (FTIR; Perkin Elmer, Waltham, MA, USA) and X-ray photoelectron spectroscopy (XPS; Thermo Fisher Scientific, Waltham, MA, USA).

### 3.4. Electrochemical Measurements

Cyclic voltammetry (CV), galvanostatic charge–discharge (GCD), and electrochemical impedance spectroscopy [72] measurements of the samples were analyzed using an electrochemical Zive SP2 instrument. The electrochemical measurements were performed in a three-electrode system with the active material, deposited Ni foam (or current collector), used as a working electrode, Ag/AgCl (3 M KCl saturated) used as a reference electrode, and a platinum coil used as a counter electrode with 2 M KOH aqueous electrolyte. The working electrodes of the materials were prepared by mixing 80% of the active material with 10% carbon black and 10% PVDF with NMP solvent to make a slurry. After grinding the slurry in a mortar and pestle, the homogeneous slurry was drop-casted on well-washed Ni foam with an area of 1 × 1 cm^2^ and dried at 60 °C for 12 h. The Ni foams had an active mass of 2 mg/cm^2^ and were used as a working electrode for the electrochemical tests. The CV and GCD measurements of the samples were recorded in the potential windows of 0 to 0.5 V at different scan rates and with current densities varying from 0 to 0.45 V. Moreover, EIS measurements were carried out on Nyquist plots with a range of 100 kHz to 0.01 Hz. The areal capacitance values were calculated from the GCD curve by using the following equation:(5)$$C=\frac{2I\times \Delta t}{A\times \Delta v}$$
where *I* is the charge discharge current (*A*), Δ*t* is the discharge time (s), *A* is the area of the electrode, and Δ*v* is the difference in the potential window (V).

### 3.5. Fabrication of Asymmetric Supercapacitors Device

To demonstrate the practical electrochemical performance of HCuS@3D-CNFs, the asymmetric supercapacitor [41] was assembled using HCuS@3D-CNFs as a positive electrode and biomass-derived activated carbon [42] as a negative electrode. The physicochemical properties of the BAC were characterized in our previous report [48]. Prior to the device assembly, the electrochemical performances of the negative electrode were also evaluated under a three-electrode system. The optimum mass ratio for the positive and negative electrodes can be calculated by the equation of charge balance theory based on their specific capacitance by the formula given below:(6)$$\frac{m^{+}}{m^{-}}=\frac{Cs^{-}\times \Delta V^{-}}{Cs^{+}\times \Delta V^{+}}$$
where *m*, *Cs,* and Δ*V* are the active mass values on the surface of the current collector, the specific capacitance, and the working potential window of positive and negative electrodes, respectively. The electrochemical performance of the assembled cell (HCuS@3D-CNFs//BAC) was carried out using CV, GCD, and EIS under a wide potential window range (0 V–1.5 V) in a two-electrode system using 2 M KOH electrolyte-shocked Whatmann filter paper as a separator. The areal energy density (E) and powder density (P) of the cell were calculated according to the equations given below:(7)$$E=\frac{C\times {\left(\Delta V\right)}^{2}}{7.2}$$
(8)$$P\;=\frac{E\;\times 3600}{\Delta t}$$
where *C* is the areal capacitance (F/cm^2^) of the asymmetric cell, Δ*V* is the voltage difference, Δ*t* is the discharge time (s), E is the energy density (mWh/cm^2^), and P is the power density (mW/cm^2^).

## 4. Conclusions

In summary, we have successfully synthesized hollow CuS (HCuS) spheres integrated with 3D-CNF nanoarchitectonics via electrospinning, NaBH4 treatment, and the hydrothermal method. The 3D-CNF networks provided conductive support for the growth of HCuS and prevented the agglomeration of spheres. Compared to HCuS, the as-synthesized HCuS@3D-CNF nanoarchitectonics delivered outstanding specific capacitance and rate capability. Moreover, an asymmetric device was assembled using HCuS@3D-CNFs and BAC as positive and negative electrode materials, respectively. The as-assembled HCuS@3D-CNFs//BAC asymmetric device delivered outstanding areal energy density of 0.15 mWh/cm^2^ at the power density of 2.92 mW/cm^2^ and admirable cyclic durability after 5000 GCD cycles. The presented work provides a promising route for the synthesis of HCuS@3D-CNF nanoarchitectonics for high-performance supercapacitor applications.

## Figures and Tables

**Figure 1 ijms-24-09685-f001:**
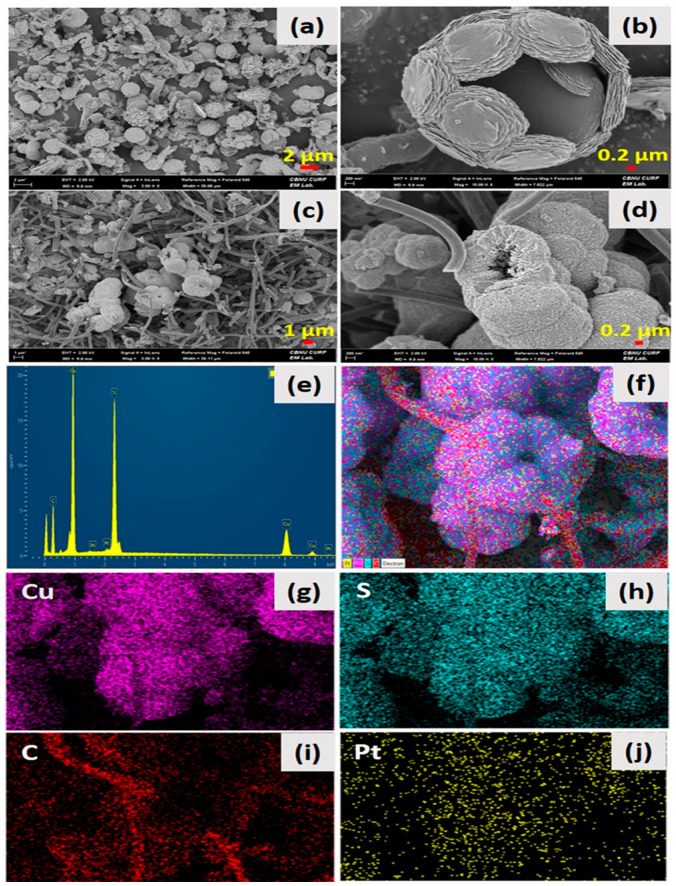
Low abd high magnification FE-SEM images: of pristine HCuS spheres (**a**,**b**) HCuS@3D-CNF nanoararchitectonics (**c**,**d**) EDS spectra (**e**) and color mapping of HCuS@3D-CNF nanoararchitectonics (**f**–**j**), respectively.

**Figure 2 ijms-24-09685-f002:**
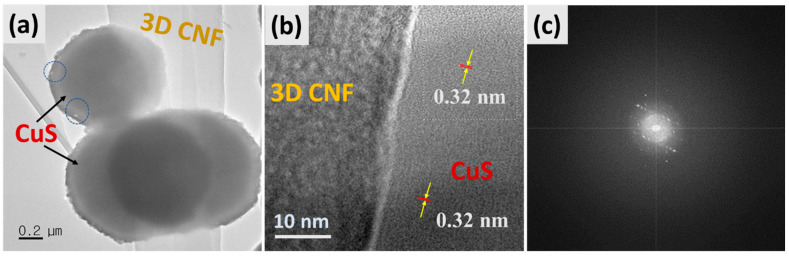
TEM images of HCuS@3D-CNF nanoararchitectonics (**a**), HR-TEM (**b**), and SAED pattern (**c**).

**Figure 3 ijms-24-09685-f003:**
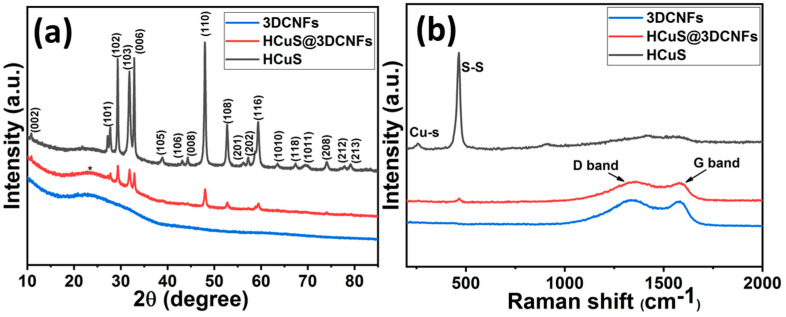
XRD patterns (**a**) and Raman spectra (**b**) of individual materials.

**Figure 4 ijms-24-09685-f004:**
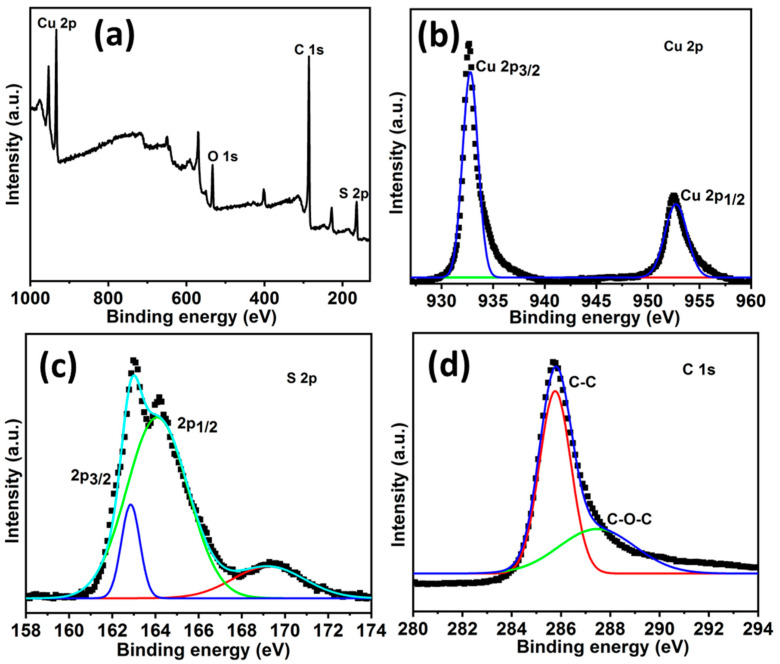
XPS spectra of HCuS@3D-CNF nanoararchitectonics: wide-range survey (**a**), the Cu 2p region of HCuS@3D-CNFs (**b**), the S 2p region of HCuS@3D-CNFs (**c**), and the C 1s region of HCuS@3D-CNFs (**d**).

**Figure 5 ijms-24-09685-f005:**
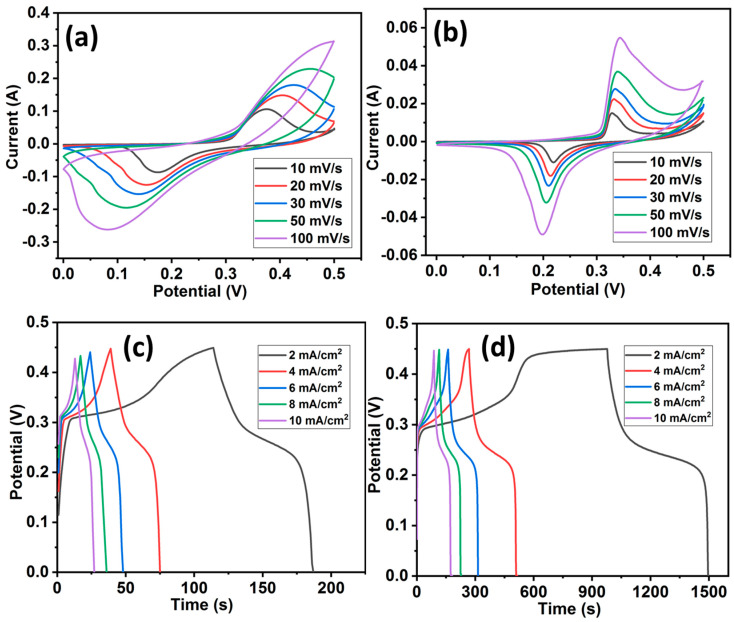
Cyclic voltammetry curves of HCuS (**a**) and HCuS@3D-CNFs (**b**); and GCD curves of HCuS (**c**) and HCuS@3D-CNFs (**d**).

**Figure 6 ijms-24-09685-f006:**
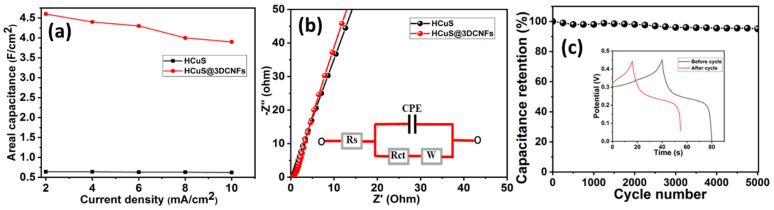
Areal capacitance vs. current density (**a**); EIS Nyquist plots with equivalent circuit model (inset) (**b**); cyclic stability test of HCuS@3D-CNFs (**c**).

**Figure 7 ijms-24-09685-f007:**
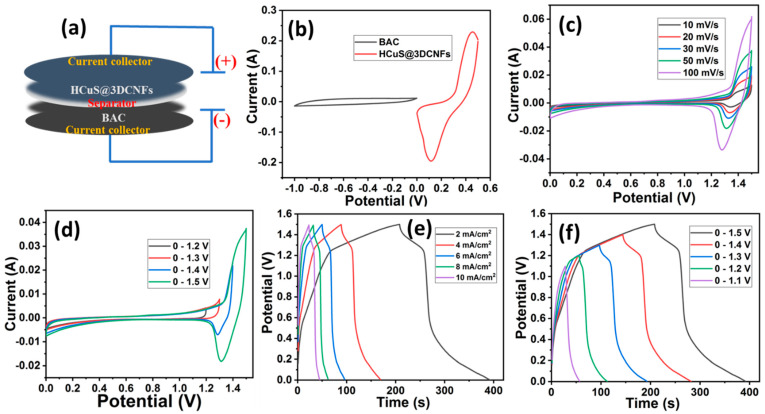
Schematic illustration of assembled electrochemical cell HCuS@3D-CNFs//BAC (**a**), comparative CV curves of HCuS@3D-CNFs (positive electrode) and BAC (negative electrode) at 50 mV/s in 2M KOH electrolyte (**b**), CV curves of HCuS@3D-CNFs//BAC measured at various scan rates (**c**), CV curves of HCuS@3D-CNFs//BAC measured over different potential windows at a scan rate of 50 mV/s (**d**), GCD curve of HCuS@3D-CNFs//BAC measured at different current densities and potential windows (**e**,**f**).

**Figure 8 ijms-24-09685-f008:**
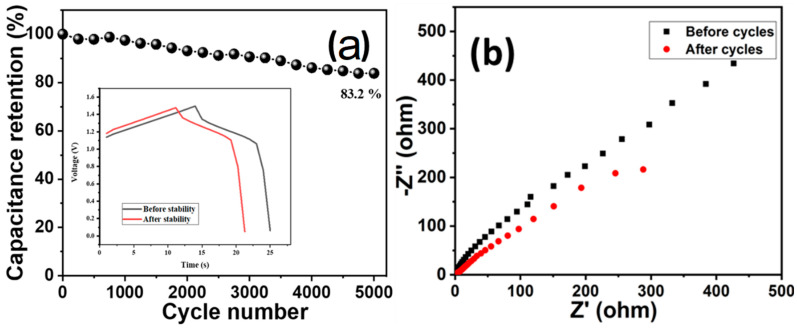
Cyclic stability (**a**) and EIS (before and after cycle) of asymmetric device (**b**).

**Figure 9 ijms-24-09685-f009:**
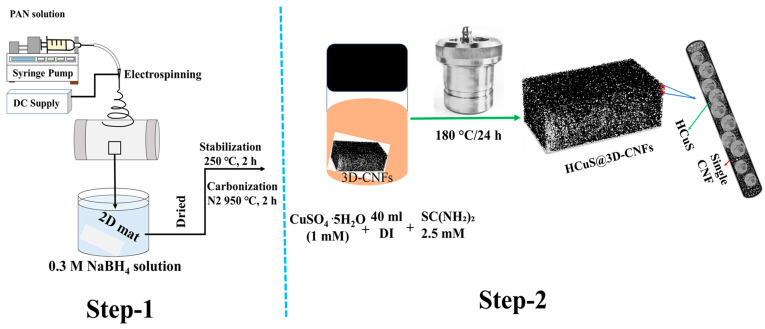
Schematic illustration showing overall synthesis method for HCuS@3D-CNF nanoararchitectonics.

**Table 1 ijms-24-09685-t001:** Comparison with the specific capacitance of related material in the literature.

Electrode Materials	Electrolyte Solution	Capacitance	Energy Density/Power Density	Stability	Ref.
CMS@GC composite	3 M KOH	174 F/g	5.5 Wh/Kg, 48 W/Kg	78.94% after 2500 cycles	[61]
MoS_2_-Bi_2_S_3_@CNT composite	2 M KOH	1338 F/g at 10 mV/s	-	60% at 0.5 A/g over 2000 cycles	[62]
Cu_7_S_4_/C composite	1 M H_2_SO_4_	321.9 F/g at 0.5 A/g	-	78.1% at 3000 cycles	[63]
MoS_2_/CB-C	1 M Na_2_SO_4_	333.5 F/g at 1 A/g	7.6 Wh/Kg	81.8% at 7000 cycles	[64]
ZnS/G nanocomposite	6 M KOH	197.1 F/g at 5 mV/s	-	89.2% after 2000 cycles	[65]
Carbon–MoS_2_–-carbon nanoplates	1 M LiSO_4_	0.12 F/cm^2^ at 0.1 A/g	-	85% after 3000 cycles	[66]
MoS_2_@CNT/RGO	1 M H_2_SO_4_	129 mF/cm^2^ at 0.1 mA/g	-	94.7% after 10,000 cycles	[67]
MoS_2_@CNT heterostructure	1 M Na_2_SO_4_	131 mF/cm^2^ at 5 mV/s	-	97.6% after 2500 cycles	[68]
HCuS@3D-CNF composite	2 M KOH	4.6 F/cm^2^ at 2 mA/cm^2^	0.15 mWh/cm^2^	~95% at 5000 cycles	This work

## Data Availability

Not applicable.

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
