# Peer review of "Nanoarchitectonics of Three-Dimensional Carbon Nanofiber-Supported Hollow Copper Sulfide Spheres for Asymmetric Supercapacitor Applications"

_ijms, 2023, doi:10.3390/ijms24119685_

Round 1

Reviewer 1 Report

Although tis work has certainly routine features, valid important data area actually included. Publication of these data in public journal media would have certain good contributions in the related research communities. From this positive viewpoints, I may recommend publication of this work in International Journal of Molecular Sciences. However, more impacts have to be somehow added by revisions to remove routine impression and add innovative impression. I suggest several revisions on rather general points. Please see below.

1) Because the research targets are hot ones. Superiorities and advantageous features of this system have to be discussed upon comparisons over the related materials system reported in the past literatures. Detailed and quantitative discussions have to be done for reasons and mechanisms od the observed superior properties.

2) Inclusion of a new conceptual term in the title often changes impression of the work. I may suggest use of an emerging conceptual term, nanoarchitectonics, in the title (as post-nanotechnology concept, see https://pubs.rsc.org/en/content/articlelanding/2021/nh/d0nh00680g). For example, the title like ... Nanoarchitectonics of Three-Dimensional Carbon Nanofibers Supported Hollow Copper Sulfide Spheres for Asymmetric Supercapacitor Applications ... may sound more innovative.

3) Figure 1 has to be improved. The current ones mainly show processes with apparatus cartoons where scientific information is less. Please add more chemical structures to explain what kinds of the materials are prepared in these processes.

4) Descriptions on conclusive sections are weak. Please include more descriptions for future perspectives wit the obtained results. It gives advanced impression of meaning of this work.

5) In the title page, "and" in the author list looks strange.

Author Response

Reviewer 1# Although this work has certainly routine features, valid important data area included. Publication of these data in public journal media would have certain good contributions in the related research communities. From this positive viewpoint, I may recommend publication of this work in International Journal of Molecular Sciences. However, more impacts have to be somehow added by revisions to remove routine impression and add innovative impression. I suggest several revisions on rather general points. Please see below.

We are pleased to reviewers for the positive and constructive feedbacks, which are helpful to improve the quality of our manuscript. We have revised the manuscript according to the reviewer’s comments

  • Because the research targets are hot ones. Superiorities and advantageous features of this system have to be discussed upon comparisons over the related materials system reported in the past literatures. Detailed and quantitative discussions have to be done for reasons and mechanisms of the observed superior properties.

Response: Thank you for your valuable suggestion. The related materials are compared in (Table. 1) in revised manuscript. Moreover, advantages of the 3D CNFs with hollow copper sulfide composite are also mentioned in the revised manuscript.

  • Inclusion of a new conceptual term in the title often changes impression of the work. I may suggest use of an emerging conceptual term, nanoarchitectonics, in the title (as post-nanotechnology concept, see https://pubs.rsc.org/en/content/articlelanding/2021/nh/d0nh00680g). For example, the title like ... Nanoarchitectonics of Three-Dimensional Carbon Nanofibers Supported Hollow Copper Sulfide Spheres for Asymmetric Supercapacitor Applications ... may sound more innovative.

Response: Thank you for your valuable suggestion. As-per your suggestion, wee have changed the title of the manuscript as follow “Nanoarchitectonics of Three-Dimensional Carbon Nanofibers Supported Hollow Copper Sulfide Spheres for Asymmetric Supercapacitor Applications”

Figure 1 has to be improved. The current ones mainly show processes with apparatus cartoons where scientific information is less. Please add more chemical structures to explain what kinds of the materials are prepared in these processes.

Response: Thank you for your suggestion. As per your suggestion we have improved the schematic diagram of Fig. 1. along with chemical structures of synthesized materials. 

Figure 1. Schematic illustration of the synthesis method for HCuS@3D-CNFs composite

 4) Descriptions on conclusive sections are weak. Please include more descriptions for future perspectives wit the obtained results. It gives advanced impression of meaning of this work.

Response: Thank you for your suggestion. The conclusions section have been revised and incorporate in the revised version of the manuscript.

  • In the title page, "and" in the author list looks strange

Response: We apologize for our mistake. The mistakes have been corrected in the revised version of the manuscript.

Reviewer 2 Report

I have reviewed the submitted paper entitled “Three-Dimensional Carbon Nanofibers Supported Hollow Copper Sulfide Spheres for Asymmetric Supercapacitor Applications” to Nanomaterials/MDPI. In recent decades, energy storage systems such as rechargeable batteries and hybrid supercapacitors have received great attention, especially in industries like hybrid electric vehicles and smart portable electronics. Various transition metal oxides and hydroxides have been researched as potential electrodes for batteries and supercapacitors due to their excellent specific capacitance derived from faradaic/electrostatic reactions on the surface. In their submitted work, the authors used hollow copper sulfide spheres HCuS@3DCNFs versus activated carbon electrodes and validated their studies in a single-cell configuration and device performance. The synthesis of copper sulfide and its composites with carbon-based materials are significant areas of research.

The work is well presented and fairly well organized, and the manuscript is fine with good electrochemical characterization. The work in its present form is publishable but needs some revisions before rendering a final decision. 

The following points need to be considered.

·         What is the advantage of converting 2D PAN electrospun into #DCNFs while amorphous/porous activated carbon-based copper sulfide composites have been elegantly used? This must be detailed in the last paragraph of the introduction.

·         Arguably, a few more papers report on CUS having different morphologies.  The researchers have also used acetylene black CuS nanosheet composites through a solvothermal process.

·         What is the area that the material has been coated on Ni Foam?

·         Page 7, line 183 – polycrystalline?

·         Please prove the equations/mechanisms corresponding to charge/discharge processes.

·         Why in Fig. 6d, the charge curve has been potential limited?

·         Figures 8 d, e, and f: In d CV curve not showing any redox reaction occurring at 0.2 V while CD curves have pronounced plateau-like a curve in the 0.2 V region. Justify.

·         Figure 8b why the AC curve is so tiny and not showing a rectangular shape.

·         Compare the obtained asymmetric supercapacitors with the reported counterpart of metal oxides such as (doi.org/10.1021/acsami.0c13755) and benchmark it for the readers to get a sense of Sulfides.

·         Strictly speaking, the drop in cycling stability over 5000 losing more than 20% of its initial capacity is not recommended. Please comment.

·         Why there is no pronounced CV peak distinguished for copper and sulfide?

·     What is the desirable electrochemical practical performance for the constructed device?

English is OK, a few minor edits and syntax errors will fix.

Author Response

Reviewer2# I have reviewed the submitted paper entitled “Three-Dimensional Carbon Nanofibers Supported Hollow Copper Sulfide Spheres for Asymmetric Supercapacitor Applications” to Nanomaterials/MDPI. In recent decades, energy storage systems such as rechargeable batteries and hybrid supercapacitors have received great attention, especially in industries like hybrid electric vehicles and smart portable electronics. Various transition metal oxides and hydroxides have been researched as potential electrodes for batteries and supercapacitors due to their excellent specific capacitance derived from faradaic/electrostatic reactions on the surface. In their submitted work, the authors used hollow copper sulfide spheres HCuS@3DCNFs versus activated carbon electrodes and validated their studies in a single-cell configuration and device performance. The synthesis of copper sulfide and its composites with carbon-based materials are significant areas of research.

The work is well presented and fairly well organized, and the manuscript is fine with good electrochemical characterization. The work in its present form is publishable but needs some revisions before rendering a final decision. 

The following points need to be considered.

We thank the reviewer for positive and constructive feedback. Your valuable comments and suggestions are helpful to improve the quality of work presented in this manuscript.

 What is the advantage of converting 2D PAN electrospun into #DCNFs while amorphous/porous activated carbon-based copper sulfide composites have been elegantly used? This must be detailed in the last paragraph of the introduction.

Response: Thank you for your valuable comment. With compared to 2D carbon nanofibers, 3D networks of CNFs have gained much interest in supercapacitor application due to their noteworthy characteristics such as easy synthesis, meso/microporosity, and high electrolyte ion accessibility and diffusion during the electrochemical test. These advantages have been also included in the revised manuscript.

  • Arguably, a few more papers report on CUS having different morphologies.  The researchers have also used acetylene black CuS nanosheet composites through a solvothermal process.

Response: Thank you for your suggestion. More reported papers on CuS with different morphologies are also included in introduction part.

  • What is the area that the material has been coated on Ni Foam?

Response: We have deposited the electroactive materials over the Nickel foam with an area of 1*1 cm2.

  Page 7, line 183 – polycrystalline?

Response: We apologized for our mistake. The mistake has been corrected in the revised manuscript.

Please prove the equations/mechanisms corresponding to charge/discharge processes.

Response: Thank you for your suggestion. We provide the eq.s corresponding to the oxidation reduction process on CuS and CuS/C composite during the electrochemical process. The following eq.s are also provided in the revised manuscript.

Reduction process:

CuS + OH- → CuSOH + e                                                                           (1)

CuSOH + OH- → CuSO + H2O + e-                                                              (2)

Oxidation process:

CuSO + H2O + e- → CuSOH + OH-                                                              (3)

CuSOH + e- → CuS + OH-                                                                             (4)

  Why in Fig. 6d, the charge curve has been potential limited?

Response: Thank you for your queries. Compared to CV, GCD has usually much smaller current density, less polarization will occur during charging, thus the high voltage cannot be realized.

 Figures 8 d, e, and f: In d CV curve not showing any redox reaction occurring at 0.2 V while CD curves have pronounced plateau-like a curve in the 0.2 V region. Justify.

Response: Thank you for your queries. In three-electrode setup, the redox potential appeased near at 0.3 and 00.2 V, however, in device relocation of redox potential appeared at 1.2V, this might be due to the effect negative electrode. The negative electrode drags the electron electrons to the more negative terminals. Therefore, the redox potential appeared at 1.2 V.

 Figure 8b why the AC curve is so tiny and not showing a rectangular shape.

Response: Thank you for your queries. Here, we have plotted the CV curves of two different materials are in the same fig. or wide potential window, which have different capacitive behavior (i. e. Double -layer capacitors and Pseudocapacitors). So, it looks tiny.

Compare the obtained asymmetric supercapacitors with the reported counterpart of metal oxides such as (doi.org/10.1021/acsami.0c13755) and benchmark it for the readers to get a sense of Sulfides.

Response: Thank you so much for your valuable suggestion. TMSs have growing attention for supercapacitor electrode due to their controlled shapes, low electronegativity, complex valence of sulfur, suitable potential window, better electrical conductivity compared to their correspond oxides, ample electrochemical redox sites, low cost, and environmental geniality.. This referred research article has been incorporated in the revised version of manuscript.

Strictly speaking, the drop in cycling stability over 5000 losing more than 20% of its initial capacity is not recommended. Please comment.

Response: We have retested the stability of the device and incorporated in the revised version of the manuscript. The as-assembled device possessed % capacitance retention after 5000 GCD cycles which is incorporated in the revised manuscript.

Why there is no pronounced CV peak distinguished for copper and sulfide?

Response: Thank you for your queries. During electrochemical process, metallic counterparts only take part in the oxidation and reduction process, whereas S possesses nonmetallic characteristic, which do not participate in the redox reaction however, it increases the conductivity of the materials. Therefore, the distinct peaks for Cu and S cannot been seen during redox reaction.  

Reduction process:

CuS + OH- → CuSOH + e                                                                           (1)

CuSOH + OH- → CuSO + H2O + e-                                                              (2)

Oxidation process:

CuSO + H2O + e- → CuSOH + OH-                                                              (3)

CuSOH + e- → CuS + OH-                                                                             (4)

What is the desirable electrochemical practical performance for the constructed device?

Response: Thank you for your queries. The desirable practical performance of the constructed device is to enhance the specific capacitance and energy density as compared to another 2D carbonaceous material.

Round 2

Reviewer 1 Report

Replies and revisions are fine. The revised version becomes accept.

Reviewer 2 Report

I went through the author's responses and the revised part of the manuscript. In this reviewer's opinion, it appears to be reasonably well addressed. Therefore, the revised version is suitable for publication.